# $Pb^{2+}$ biosorption from aqueous solutions by live and dead biosorbents of the hydrocarbon-degrading strain *Rhodococcus* sp. HX-2

Xin Hu[1][○], Jiachang Cao[1][○], Hanyu Yang[2], Dahui Li[1], Yue Qiao[1], Jialin Zhao[1], Zhixia Zhang[1], Lei Huang[1]*

1 College of Chemistry and Chemical Engineering, Tianjin Key Laboratory of Drug Targeting and Bioimaging, Tianjin Key Laboratory of Organic Solar Cells and Photochemical Conversion, Tianjin University of Technology, Tianjin, China, 2 College of Management Science and Engineering, Capital University of Economics and Business, Beijing, China

○ These authors contributed equally to this work.
* huanglei@tjut.edu.cn

**Data Availability Statement:** All relevant data are within the manuscript and its Supporting Information files.

## Abstract

In this study, the $Pb^{2+}$ biosorption potential of live and dead biosorbents of the hydrocarbon-degrading strain *Rhodococcus* sp. HX-2 was analyzed. Optimal biosorption conditions were determined via single factor optimization, which were as follows: temperature, 25°C; pH, 5.0, and biosorbent dose, 0.75 g $L^{-1}$. A response surface software (Design Expert 10.0) was used to analyze optimal biosorption conditions. The biosorption data for live and dead biosorbents were suitable for the Freundlich model at a $Pb^{2+}$ concentration of 200 mg $L^{-1}$. At this same concentration, the maximum biosorption capacity was 88.74 mg $g^{-1}$ (0.428 mmol $g^{-1}$) for live biosorbents and 125.5 mg $g^{-1}$ (0.606 mmol $g^{-1}$) for dead biosorbents. Moreover, in comparison with the pseudo-first-order model, the pseudo-second-order model seemed better to depict the biosorption process. Dead biosorbents seemed to have lower binding strength than live biosorbents, showing a higher desorption capacity at pH 1.0. The order of influence of competitive metal ions on $Pb^{2+}$ adsorption was $Cu^{2+} > Cd^{2+} > Ni^+$. Fourier-transform infrared spectroscopy analyses revealed that several functional groups were involved in the biosorption process of dead biosorbents. Scanning electron microscopy showed that $Pb^{2+}$ attached to the surface of dead biosorbents more readily than on the surface of live biosorbents, whereas transmission electron microscopy confirmed the transfer of biosorbed $Pb^{2+}$ into the cells in the case of both live and dead biosorbents. It can thus be concluded that dead biosorbents are better than live biosorbents for $Pb^{2+}$ biosorption, and they can accordingly be used for wastewater treatment.

## Introduction

Water pollution caused by heavy metals is an increasingly serious global issue. Heavy metals accumulate through the food chain even at very low concentrations, posing an enormous

**Funding:** This work was supported by the Nankai laboratory, the National Natural Science Foundation of China (Grant No. 21777113), the Natural Science Foundation of Tianjin City (Grant No. 15JCQNJC08800), the National Undergraduate Training Programs for Innovation and Entrepreneurship (No. 201610060037 and 201710060027), and the Training Project of the Innovation Team of Colleges and Universities in Tianjin. And the grant recipient is corresponding author: Lei Huang. The funders had no role in study design, data collection and analysis, decision to publish, or preparation of the manuscript.

**Competing interests:** The authors have declared that no competing interests exist.

threat to humans, animals, and the environment [1–4]. Cadmium, chromium, copper, lead, and zinc, all of which are toxic heavy metals, are widely used across various industries [5]. Toxic heavy metal ions can emerge from a wide range of sources and have become the major pollutants in water resources [6]. Lead pollution specifically is a major concern, as lead accumulation in the environment dates back to industrialization [7]. There are various sources of lead such as gasoline, batteries, mining, electroplating, lead smelting, photographic materials, and explosives [8]. Considering its toxicity and diffusion in the environment, lead is the primary target of heavy metal management [9]. In China, lead production has reached 135 million tons per year and the consumption has exceeded 80 million tons since 2004 [10]. Therefore, eradicating lead contamination is very urgent and pivotal.

Toxic metals in wastewater are usually removed using physicochemical methods such as precipitation, membrane technology, ion exchange, and activated carbon adsorption [11]. However, these techniques are either expensive for processing secondary byproducts or ineffective at high metal ion concentrations. Sometimes, complex synthesis of exchangers is essential [12–16]. These methods have multiple disadvantages, including high cost, low efficiency, and re-contamination due to the generation of toxic matter [17–19]. Bacteria are comparatively more effective for toxic metal adsorption, specifically at low concentrations of heavy metals in solutions [20–22].

A significant characteristic of biosorbents is that they can be either live or dead. Most studies on metal eradication involve the use of dead biosorbents as the preferred alternative so as to reduce complexity [23]. Paul et al. reported that autoclaving a bacterial biosorbent strengthened its ability to biosorb heavy metal ions [24]; this could be attributed to the degradation of the cell wall, exposing a potential binding site for a higher number of metal ions. Dead biosorbents have multiple advantages in comparison with their live counterparts, such as high efficiency, no requirement of growth media or nutrients, reduced waste sludge generation, and low cost [25]. However, live biosorbents have their own set of advantages. They can transfer adsorbed heavy metals into cells and alter the state of heavy metal ions to reduce their toxicity [26]; moreover, live biosorbents can more efficiently eradicate heavy metals at a low concentration [27]. However, only a few studies have compared the ability of live and dead biosorbents for toxic heavy metal biosorption.

In this study, our aim was to compare the $Pb^{2+}$ biosorption potential of live and dead biosorbents of *Rhodococcus* sp. HX-2 in aqueous solutions. Biosorbent dosage, pH, initial metal concentration, and contact time were considered to influence $Pb^{2+}$ biosorption. The biosorption process of live and dead biosorbents are described using different isotherm and kinetic models. Desorption experiments were performed to determine the ability to release metal ions and recover biosorbents, while competitive biosorption and Fourier-transform infrared spectroscopy (FT-IR) were used to identify possible binding sites and functional groups of live and dead biosorbents. Scanning electron microscopy (SEM) and transmission electron microscopy (TEM) were respectively used to determine whether $Pb^{2+}$ was adsorbed onto the surface of the biosorbents and transferred intracellularly by the biosorbents.

## Materials and methods

### Culture media

Four different media were used: Luria–Bertani medium (10 g $L^{-1}$ peptone, 5 g $L^{-1}$ yeast powder, 5 g $L^{-1}$ NaCl, pH 7.2), minimal salt medium [1.5 g $L^{-1}$ $Na_2HPO_4$, 3.48 g $L^{-1}$ $KH_2PO_4$, 4 g $L^{-1}$ $(NH_4)_2SO_4$, 0.7 g $L^{-1}$ $MgSO_4 \cdot 7H_2O$, 0.01 g $L^{-1}$ yeast powder, pH 7.2), LMM (low phosphate mineral medium) medium [0.1 g $L^{-1}$ $Na_2HPO_4$, 0.1 g $L^{-1}$ $KH_2PO_4$, 0.5 g $L^{-1}$ $NH_4NO_3$, 0.5 g $L^{-1}$ $(NH_4)_2SO_4$, 0.2 g $L^{-1}$ $MgSO_4$, 0.02 g $L^{-1}$ $CaCl_2$, 0.002 g $L^{-1}$ $FeCl_2$, 0.002 g $L^{-1}$ $MnSO_4$,

pH 6.5) [28], and saline medium (9 g L$^{-1}$ NaCl). The seed medium was supplemented with 2% ethanol. All media were prepared using distilled water. Luria–Bertani solid media contained 20 g L$^{-1}$ agar. Several filter-sterilized heavy metal salt stock solutions [Cu(NO$_3$)$_2$, Pb(NO$_3$)$_2$, Cd(NO$_3$)$_2$, NiNO$_3$] were separately prepared or mix together in distilled water, and they were used at varied final concentrations. All media were sterilized at 121˚C for 30 min and cooled to room temperature, the stock solutions were then added.

## Preparation of biosorbents and Pb$^{2+}$ stock solution

*Rhodococcus* sp. HX-2, a hydrocarbon-degrading strain, was isolated from oil-contaminated soil in Xinjiang Oil Field, China (43.56–45.62N, 84.13–85.08E). The study did not involve private land, protected land, endangered or protected species. No specific permissions were required for these locations/activities. This strain was inoculated into sterile Luria–Bertani medium and cultured in a thermostatic shaker (200 rpm) at 25˚C. Subsequently, the cells were harvested in the logarithmic growth phase by centrifugation at 10,000 ×*g* for 10 min at 4˚C. They were then washed three times with 150 mM NaCl. Live biosorbents were obtained by harvesting the cells by centrifugation (11,000 ×*g*, 15 min), while dead biosorbents were obtained by autoclaving the cells at 121˚C for 30 min, as suggested by Cheng et al. [29]. After washing with distilled water three times, live and dead biosorbents were used for Pb$^{2+}$ biosorption.

## Susceptibility of *Rhodococcus* sp. HX-2 to Pb$^{2+}$

The microbial susceptibility of HX-2 to Pb$^{2+}$ was detected using Pb(NO$_3$)$_2$ at various concentrations (0, 10, 20, 50, 100, 200, 300, 500 mg L$^{-1}$) with an established method [30–31]. LMM culture medium including Pb$^{2+}$ was incubated in a thermostatic shaker (200 rpm) at 25˚C for 3 days. Cell density was determined by measuring the absorbance at 600 nm (OD$_{600}$) [9]. For each heavy metal concentration, the experiments were performed in triplicate. The blank sample included Pb$^{2+}$ in LMM media, with the absorbance measured immediately post-inoculation. The cell density was determined using the following equation:

$$\alpha = \beta - \gamma \tag{1}$$

wherein $\alpha$ is the true value of sample measurement, $\beta$ is the actual value of sample measurement, and $\gamma$ is the actual value of blank measurement.

## Effects of biosorbent dose, pH, temperature, and contact time on biosorption

The effects of biosorbent dose (0.25–1.25 g L$^{-1}$), pH (3.0–7.0), temperature (10˚C–40˚C), and contact time (2.5–30 min) on Pb$^{2+}$ biosorption and removal rate by live and dead biosorbents were examined to determine optimal conditions. All samples were inoculated at 200 mg L$^{-1}$ Pb$^{2+}$. Biosorption experiments were performed at a stirring speed of 200 rpm, unless otherwise stated. At the end of the tests, residual Pb$^{2+}$ levels in the supernatant were measured using inductively coupled plasma optical emission spectroscopy after centrifugation [32]. The removal rate and biosorption capacity of Pb$^{2+}$ were measured as follows [33]:

$$q_e = \frac{(C_0 - C_e)V}{X} \quad \text{Removal rate (\%)} = \frac{C_0 - C_e}{C_0} \times 100\%$$

wherein $q_e$ is the equilibrium Pb$^{2+}$ concentration on the biosorbents (mg g$^{-1}$ dry cell); $C_0$ and

$C_e$ are the initial and residual metal concentrations (mg L$^{-1}$), respectively; $V$ and $X$ is the adsorbate volume (L) and biosorbent concentration (g dry cell L$^{-1}$), respectively.

## Box Behnken design

Box Behnken Design (BBD) is an option for balancing incomplete block designs using several evenly spaced levels to represent the response surface [34]. According to experimental result, biosorbent dose (0.5–1 g), pH (3–7), temperature (15–25°C) and contact time (5–10 min) were chosen as experiment variables, while the assessed response was biosorption capacity of Pb$^{2+}$. The four-factor and three-level BBD consisting of 29 experimental runs were used to optimize the biosorption process. S1 Table lists the actual and coded variables and their respective levels. S2 Table presents the experimental design and corresponding response data.

**Modeling and statistical analysis.** The statistical and modeling analysis were managed using a software of Design Expert 10.0 (Stat-Ease Inc., Minneapolis, MN, USA). Experimental-result were fitted to a quadratic polynomial model, which can be shown as Eq [35–36]:

$$Y = a_0 + \sum_{i=1}^{4} a_i X_i + \sum_{i=1}^{4} a_{ii} X_i^2 + \sum \sum_{i<j=1}^{4} a_{ii} X_i X_j$$

where $Xi$ and $Xj$ are independent variables; $a_0$, $ai$, $aii$ and $aij$ are regression coefficients for intercept, linear, quadratic and interaction terms, respectively.

## Effects of Pb$^{2+}$ concentration and NaCl on biosorption experiments

Live and dead biosorbents were suspended in the saline medium containing different Pb$^{2+}$ concentrations (1–300 mg L$^{-1}$). All biosorption experiments were performed under optimal conditions that were experimentally determined.

To study the effects of NaCl and betaine on biosorption experiments, live and dead biosorbents were suspended in NaCl (2%–10%). Betaine (150 mg L$^{-1}$) was added as the control. Residual metal ions were monitored at the end of the biosorption process. Other biosorption conditions were the same as those in previous experiments.

## Modeling of biosorption isotherms and adsorption kinetics

The Langmuir and Freundlich adsorption isotherms are properly fitted to describe the adsorption data for a wide extent of adsorbate concentrations.

The Langmuir model is described by the formula:

$$q_e = \frac{Q_{max} b C_e}{1 + b C_e} \tag{2}$$

where $Q_{max}$ is the maximum biosorption and $b$ represents the affinity between biosorbent and biosorbate. The reciprocal form of the equation is:

$$\frac{1}{q_e} = \frac{1}{Q_{max}} + \frac{1}{C_e b Q_{max}} \tag{3}$$

The Freundlich isotherm model is described by the following formula:

$$q_e = K C_e^{1/n} \tag{4}$$

where $K$ and *1/n* are isotherm constants.

The logarithmic form of the equation is given as:

$$\log q_e = \log K + \frac{1}{n} \log c_e \tag{5}$$

The measurement of adsorption kinetics is almost identical to the isotherm experiment. The sample to be tested is taken out at predetermined time intervals and the concentration of heavy metal ions thereof is measured. The amount of adsorption at time $t$, $q_t$ (mg g$^{-1}$), was calculated by:

$$q_t = \frac{(C_0 - C_t)V}{X} \tag{6}$$

where $C_0$ and $C_t$ (mg L$^{-1}$) are the concentrations of heavy metal ions at initial and any time, respectively. $X$ is the biosorbent dose (g) and $V$ is the volume of the solution (L).

### Effects of pH on desorption experiment and biosorbent recycling

Live and dead biosorbents loaded with Pb$^{2+}$ were obtained from earlier experiments. The biosorbent was resuspended in deionized water after three washes with distilled deionized water. As per the method reported by Li et al. [37], the pH values of the solutions were then adjusted to 1.0–7.0 with 1 M HCl. The biosorbent was then harvested by centrifugation (11,000 ×$g$, 20 min), and after 24 h, metal ions released into the supernatant were immediately detected by inductively coupled plasma optical emission spectroscopy. Desorption efficiency (%) was calculated using the following equation:

$$\text{Desorption efficiency (\%)} = \frac{D_r}{D_a} \times 100\% \tag{7}$$

wherein $Dr$ is the amount of metal ions released in the supernatant (mg) and $Da$ is the amount of metal ions initially adsorbed on the biosorbent (mg).

To regenerate the biosorbent, Pb$^{2+}$-loaded biosorbents (both live and dead) were treated with HCl (pH = 1) at 25˚C with a shaking speed of 200 rpm for 12 h. Subsequently, the biosorbent was collected by centrifugation at 11,000 ×$g$ for 10 min, and Pb$^{2+}$ concentration was determined as described earlier (inductively coupled plasma optical emission spectroscopy). The regenerated biosorbent was re-used for further biosorption, and the biosorption–desorption process was repeated for five cycles.

### Effects of coexisting ions on Pb$^{2+}$ biosorption

The effect of coexisting ions on Pb$^{2+}$ biosorption was elucidated in the presence of copper, cadmium, and nickel. Solutions containing 0.5 mmol L$^{-1}$ of each metal ion were incubated with live and dead biosorbents. The conditions for competitive biosorption were set based on the optimal parameters derived from previous results. The removal rate of Pb$^{2+}$ for different combinations was monitored using inductively coupled plasma optical emission spectroscopy.

### Characterization

FT-IR spectra of Pb$^{2+}$-loaded live and dead biosorbents were obtained using a Tensor 27 spectrometer (Bruker, Germany). The dried samples were mixed with KBr (1:100) and immediately analyzed using the spectrometer in the range of 4000–400 cm$^{-1}$ with a resolution of 4 cm$^{-1}$. The background was automatically subtracted from the sample spectra. Before and after Pb$^{2+}$ biosorption, live and dead biosorbents were vacuum dried. The samples were then gold sprayed under vacuum and low pressure, followed by observation with a scanning electron

microscope equipped with an energy dispersive spectrometer (FEI Verios 460L, USA) at 2 kV accelerating voltage. In situ high-resolution TEM images were obtained using a TALOS F200X transmission electron microscope equipped with an energy dispersive spectrometer at 200 kV. Live and dead biosorbents were resuspended in deionized water and sonicated. The crushed biosorbent was then diluted 10 times with deionized water, and 100 µL of this suspension was added dropwise on an ultrathin carbon-coated copper mesh and allowed to dry.

## Results

### Effects of Pb$^{2+}$ concentration on cell growth

The growth of HX-2 at different Pb$^{2+}$ concentrations is shown in Fig 1. HX-2 can tolerate 0–200 mg L$^{-1}$ Pb$^{2+}$; low Pb$^{2+}$ concentrations (10–100 mg L$^{-1}$) actually promote its growth, which could be attributed to heavy metal transfer into the cell. The metal accumulation first occurs in the cell, followed by further mineralization to form non-toxic metal mineralization [38–39]. At relatively low concentrations, metals such as copper and zinc are indispensable for microorganisms as they provide vital cofactors for some proteins and enzymes [40]. However,

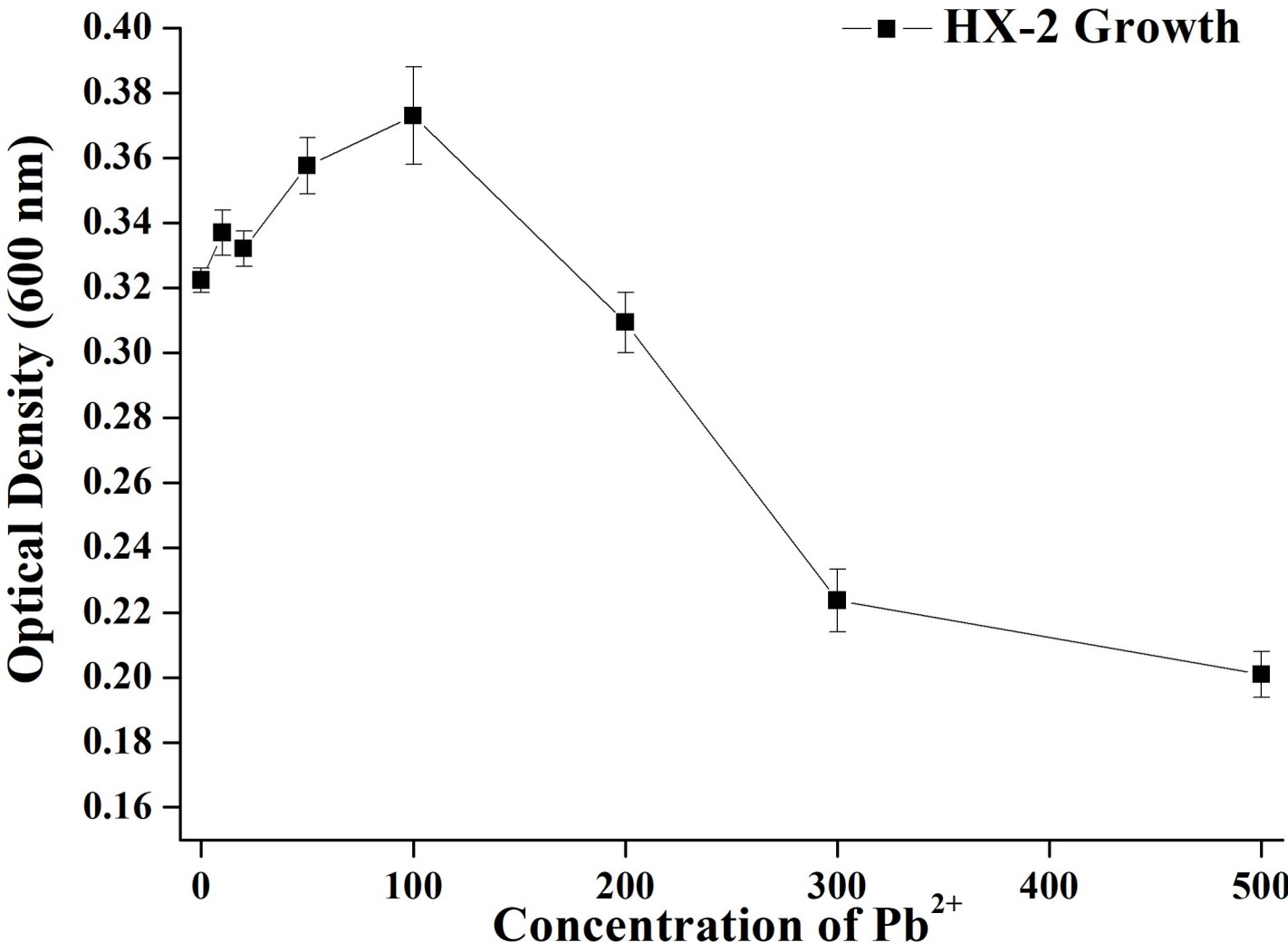

**Fig 1. Effect of Pb$^{2+}$ concentrations on the growth of strain HX-2.**

when $Pb^{2+}$ concentration exceeded 200 mg $L^{-1}$, $OD_{600}$ values rapidly decreased and reached a minimum at 500 mg $L^{-1}$ ($OD_{600}$ = 0.2), indicating that excessive $Pb^{2+}$ can markedly hinder bacterial growth. To prevent the growth from being interfered by heavy metals and to consider the maximum tolerance to $Pb^{2+}$, $Pb^{2+}$ concentration of 200 mg $L^{-1}$ was used in subsequent experiments.

## Conventional parameters

**Effects of biosorbent dose.** As the biosorbent dose increased from 0.25 to 1.25 g $L^{-1}$, the removal rate of live and dead biosorbents gradually increased too, but the biosorption capacity decreased (Fig 2A). In case of live biosorbents, the removal rate of $Pb^{2+}$ was 53.4% at 0.75 g $L^{-1}$ dosage, and the biosorption capacity was 76.9 mg $g^{-1}$, being comparable to 57.6 mg $g^{-1}$ at 1.0 g $L^{-1}$ and 50.2 mg $g^{-1}$ at 1.25 g $L^{-1}$. In case of dead biosorbents, the maximum biosorption capacity of 135.7 mg $g^{-1}$ was achieved at the lowest dose, with the removal rate being merely 32.8%. Furthermore, at a dose of 0.75 g $L^{-1}$, the biosorption capacity was 112.4 mg $g^{-1}$ and the removal rate of $Pb^{2+}$ was 76.4%; the overall biosorption effect was better than that at a dosage of 0.5 g $L^{-1}$ (biosorption capacity, 121.7 mg $g^{-1}$; removal rate of $Pb^{2+}$, 51.6%). Therefore, 0.75 g $L^{-1}$ was considered to be the optimal biosorbent dose for further testing, considering the sufficient biosorption capacity and high $Pb^{2+}$ removal rate.

**Effects of pH.** Fig 2B shows that pH had an important effect on $Pb^{2+}$ biosorption. For both live and dead biosorbents, $Pb^{2+}$ biosorption levels steadily increased with an increase in pH from 3.0 to 5.0, reaching the maximum capacity at pH = 5.0, which was 82.9 mg $g^{-1}$ and 121.9 mg $g^{-1}$, respectively.

**Effects of temperature.** As shown in Fig 2C, temperature had little effect on $Pb^{2+}$ biosorption by dead biosorbents, and the biosorption capacity was almost maintained at approximately 110 mg $g^{-1}$. However, with an increase in temperature (10˚C–30˚C), the biosorption capacity of live biosorbents increased too, reaching the highest value at 30˚C ($q_e$ = 83.4 mg $g^{-1}$).

**Effects of contact time.** The effect of contact time on the biosorption of live and dead biosorbents at an initial $Pb^{2+}$ concentration of 200 mg $L^{-1}$ is shown in Fig 2D. The shortest time required for the biosorption to reach equilibrium is depicted. In the presence of live and dead biosorbents, the contact time required for equilibration was 7.5 min, after which the $q_e$ value became almost constant. To ensure that the biosorption was balanced, the contact time was kept as 7.5 min in subsequent experiments.

To summarize, the optimal conditions for $Pb^{2+}$ removal and biosorption were as follows: biosorbent dose, 0.75 g $L^{-1}$; pH, 5.0; temperature, 30˚C for live and 20˚C for dead biosorbents; and contact time, 7.5 min. All subsequent biosorption experiments were performed under these conditions.

## Response surface analysis

**Statistical analysis and model fitting.** Optimization of $Pb^{2+}$ biosorption by dead biosorbents was performed by BBD. The design matrix and experimental response are shown in S2 Table. Multivariate regression analysis of experimental data was performed using Design Expert 10.0 software. The relationship between experimental variables and response values can be verified by a quadratic model, which can be proved to be Eq by actual factors(3):

$$Y = 190.24 - 15.08X_1 + 0.5X_2 + 0.63X_3 - 4.66X_3 - 3.8X_1X_2 - 2.03X_1X_3 + 4.68X_1X_4 - 4.36X_2X_3 + 7.33X_2X_4 + 0.43X_3X_4 - 76.47X_1^2 - 6.19X_2^2 - 5.50X_3^2 + 1.41X_4^2$$

where $Y$ is the biosorption capacity of $Pb^{2+}$; $X_1$, $X_2$, $X_3$ and $X_4$ are biosorbent dose (g), pH, temperature (˚C) and contact time (min), respectively.

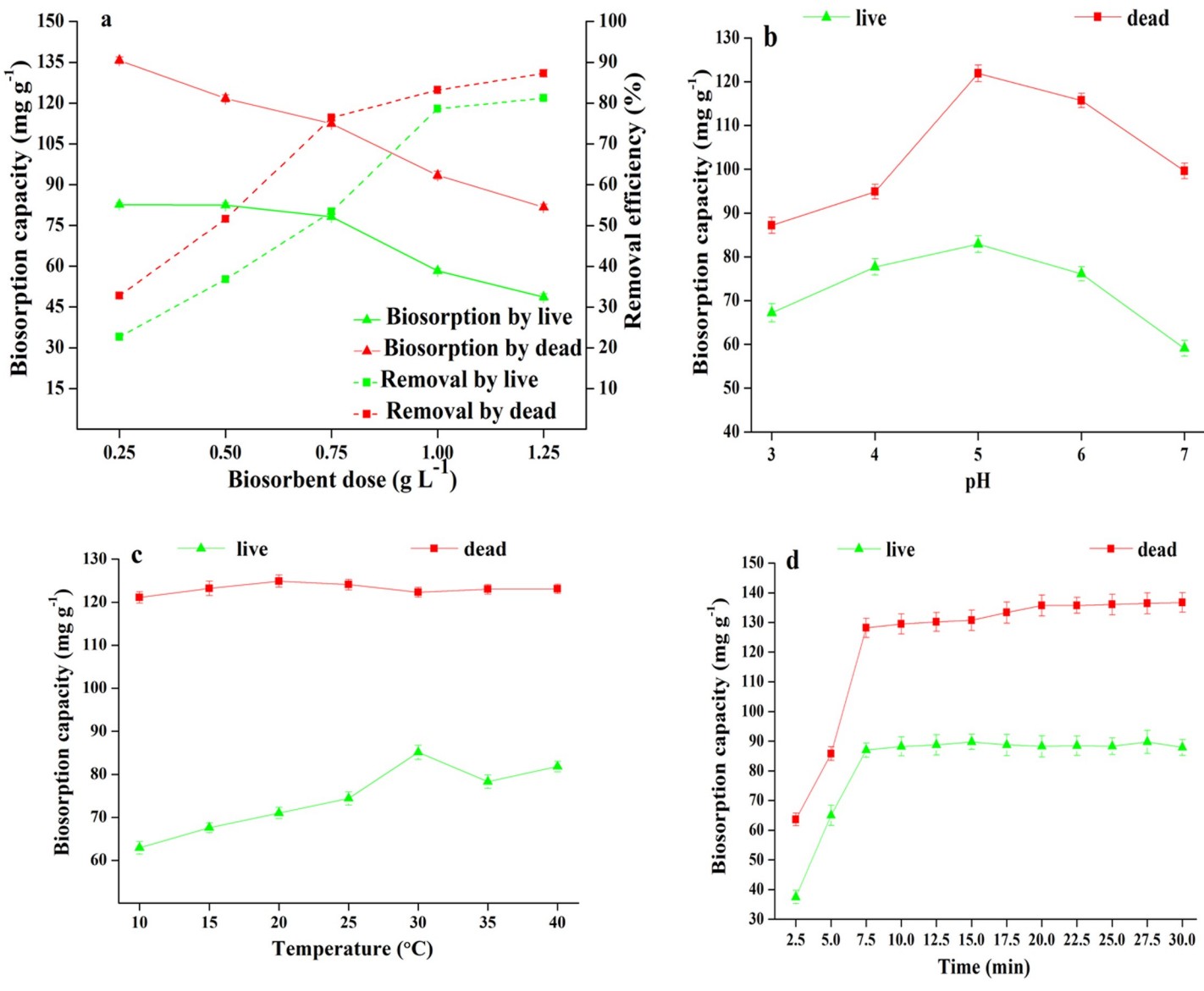

**Fig 2.** (a) Effect of biosorbent dose on biosorption capacity and removal efficiency of $Pb^{2+}$ by live and dead biosorbents. (initial $Pb^{2+}$ concentration: 200 mg $L^{-1}$; temperature: 25°C; pH: 5.0; agitation speed: 200 rpm $min^{-1}$; contact time: 30min); (b)Effect of pH on biosorption of $Pb^{2+}$ by live and dead biosorbents(initial $Pb^{2+}$ concentration: 200 mg $L^{-1}$; temperature: 25°C; agitation speed: 200 rpm $min^{-1}$; biosorbent dose: 0.75 g $L^{-1}$; contact time: 30min); (c)Effect of temperatureon biosorption of $Pb^{2+}$ by live and dead biosorbents(initial $Pb^{2+}$ concentration: 200 g $L^{-1}$; temperature: 25°C; agitation speed: 200rpm $min^{-1}$; biosorbent dose: 0.75 g $L^{-1}$; contact time: 30min, pH = 5) (d)Effect of contact time on biosorption of $Pb^{2+}$ by live and dead biosorbents (initial $Pb^{2+}$ concentration: 200 g $L^{-1}$; temperature: 20°C for dead biosorbents and 30°C for live biosorbents; agitation speed: 200rpm $min^{-1}$; biosorbent dose: 0.75 g $L^{-1}$; pH = 5).

The predicted value and the experimental value show a highly correlated straight line (S1 Fig.). This indicates that the model can effectively optimize the biosorption of $Pb^{2+}$. The sufficiency and adaptability of the model is assessed by analysis of variance (ANOVA), which provides model coefficients ($R^2$), $F$ valuesand significant probabilities. As shown in S3 Table, since the probability is less than 0.0001 and the $F$ value is 63.19, the model is very significant. "Lack of Fit" means the residual compared to the error pure error. As shown by the low $F$ value and high probability, an insignificant "Lack of Fit" indicates that the model can be used to fit experimental data. Model with a probability below 0.05 are significant. In this case, the response is

significantly effected by one linear terms ($X_1$), one interaction terms ($X_2X_4$) and one quadratic terms ($X_1^2$).

The coefficient ($R^2$ = 0.9844) agrees well with adjusted coefficient ($Adj\ R^2$ = 0.9688). And the model can explain 98.44% of experimental results. The coefficient of variation (C.V.) reflects the reliability of the experimental data. The lower value of C.V. (0.41) clearly indicates that the model is satisfactorily reproducible [41]. "Adeq Precision" describes the signal-to-noise ratio. And this ratio is 25.27 (greater than 4), indicating an sufficient signal. Therefore, the model can be used in biosorption experiments.

**Interactive effect of variables.** In order to determine the effect of the independent variable on the biosorption capacity of $Pb^{2+}$, the contour map and the response surface map were constructed according to Eq (3) (S2 Fig). S2 Fig shows that the shape of the contour map is not circular but elliptical, indicating a significant interaction between the variables [42]. Generally, higher variables including biosorbent dose, pH, temperature and contact time results in higher biosorption capacity. These factors have different differences in the contribution of biosorption capacity. It can also be seen that there are optimal conditions for maximum response.

**Optimization of biosorption process.** The software predicts that the optimal conditions are biosorbent dose 0.83g, pH 4, temperature 24.41˚C and contact time 5 min. S4 Table shows the results of the experimental verification. It can be seen that the experimental values are very close to the predicted values, indicating that the optimal conditions (biosorption dose 0.83g, pH 4, temperature 24˚C and contact time 5 min) are reliable.

## Batch experiments

**Effects of initial concentration.** As the initial metal ion concentration increased, the biosorption capacity and removal rate changed too for good (Fig 3). The $Pb^{2+}$ biosorption process of live and dead biosorbents showed a similar trend in the initial concentration range of 1–300 mg $L^{-1}$. However, the removal rate of dead biosorbents comparatively increased faster but reached equilibrium (200 mg $L^{-1}$) at the same $Pb^{2+}$ concentration. The equilibrium biosorption capacity of live biosorbents was always within the range of 0–100 mg $L^{-1}$, being almost the same as that of dead biosorbents.

**Effects of NaCl on biosorption.** NaCl concentration had no impact on the ability of dead biosorbents to adsorb heavy metals (Fig 4); this could be because dead biosorbents themselves would have denatured in the process of preparation, and thus, external parameters would have had little effect on them. On the other hand, live biosorbents continue to remain active, but considering the lack of nutrients during the biosorption process, their growth can be retarded. However, hyperosmotic conditions of the environment are bound to affect normal cellular metabolism. At lower NaCl concentrations (2% and 4%), the biosorption capacity of live biosorbents was affected by osmotic pressure (only 74.4% and 66.5% of the original biosorption capacity, respectively). At higher NaCl concentrations (6%–10%), the biosorption capacity gradually stabilized (approximately 49.2% of the original biosorption capacity).

## Isotherm studies

S3 Fig showed the data of $q_e^{-1}$ and $C_e^{-1}$ based on the Langmuir model (Eq (3)). The data of $logq_e$ versus $logC_e$, based on the Freundlich isotherms (Eq (5)) are shown in S4 Fig. A very high correlation coefficient ($R^2$) (S5 Table) can be obtained from the data of the two isotherms. The Freundlich model is more suitable than the Langmuir model for the correlation coefficient and biosorption correlation values of live and dead biosorbents.

Matching results of the Langmuir isotherm was shown in S5 Table. The maximum $Q_{max}$ value of $Pb^{2+}$ biosorption on dead biosorbentsis observed to be 187.52 mg $g^{-1}$ and while the

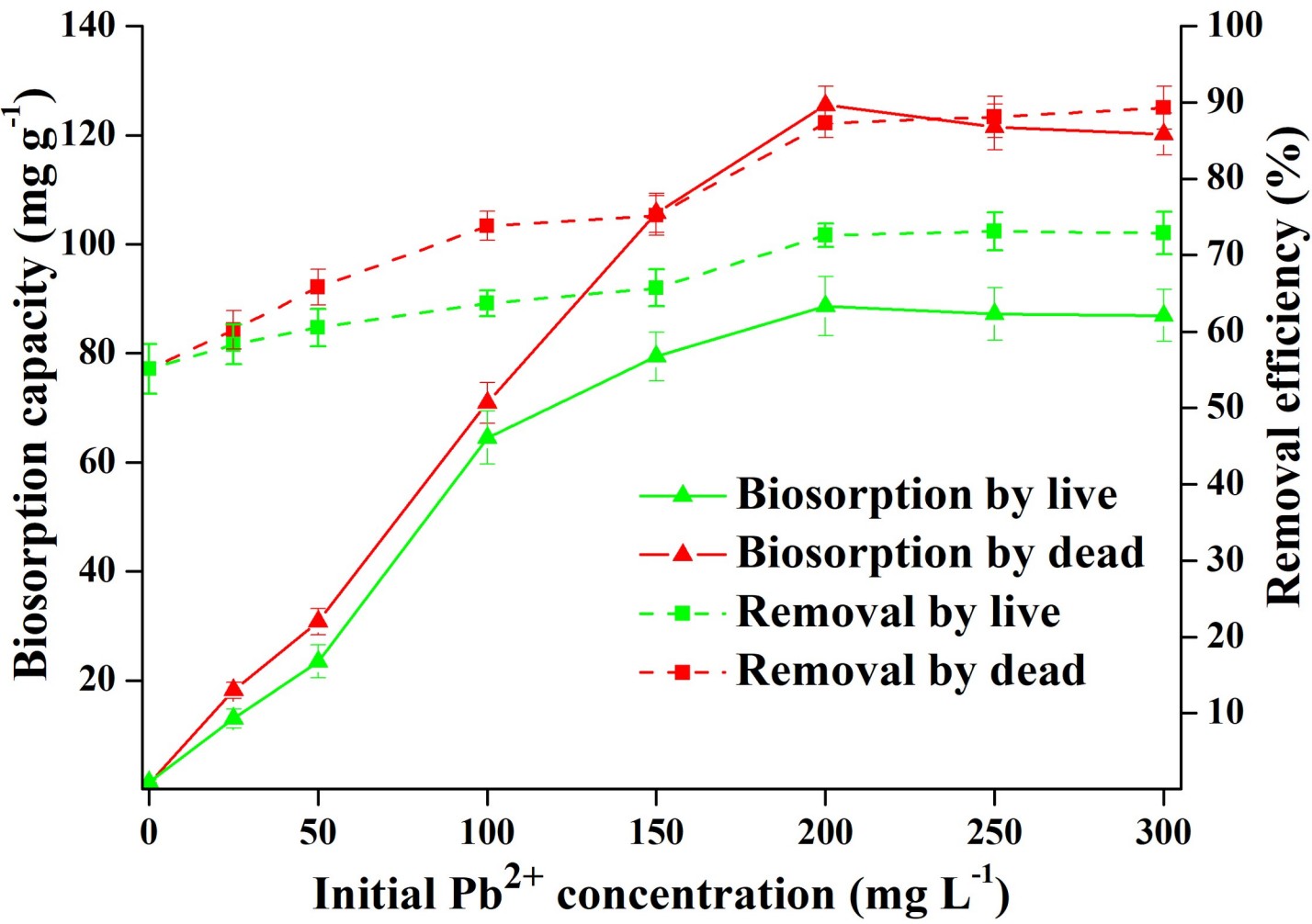

**Fig 3. Biosorption capacity and removal efficiency of $Pb^{2+}$ by live and dead biosorbents over initial concentration ranging from 1 to 300 mg $L^{-1}$.**

maximum $Q_{max}$ value was only 89.63 mg $g^{-1}$ for $Pb^{2+}$ biosorption on live biosorbents. From the datas of biosorption experiments of $Pb^{2+}$ on live and dead biosorbents, it can be seen that the biosorption of $Pb^{2+}$ by dead biosorbents has a lower *b* value, indicating that effective biosorption of $Pb^{2+}$ by dead biosorbents. The Freundlich model exhibits high *K* and *1/n* values (*n*>1) indicating high biosorption volume and biosorption strength, respectively. From the biosorption results of dead biosorbents to $Pb^{2+}$ (S5 Table), the *K* and *n* values were 19.3108 and 2.7964, respectively, indicating that dead biosorbents are better than live biosorbents.

## Biosorption kinetics

To understand the mechanism of biosorption processes such as membrane transport and chemical reactions, kinetic models can be used to explore the behavior of live and dead biosorbents. The functional groups on the cell walls of live and dead biosorbents (eg, -COOH, -OH, -$NH_2$) can interact with metal ions in many types. Three known kinetic models (first-order, second-order and intra-particle diffusion equations) can be used to analyze the biosorption behavior of biosorbents from the above two aspects [43–45]. First, the biosorption behavior of live and dead biosorbents were studied using a pseudo first-order equation (S1 Equation).

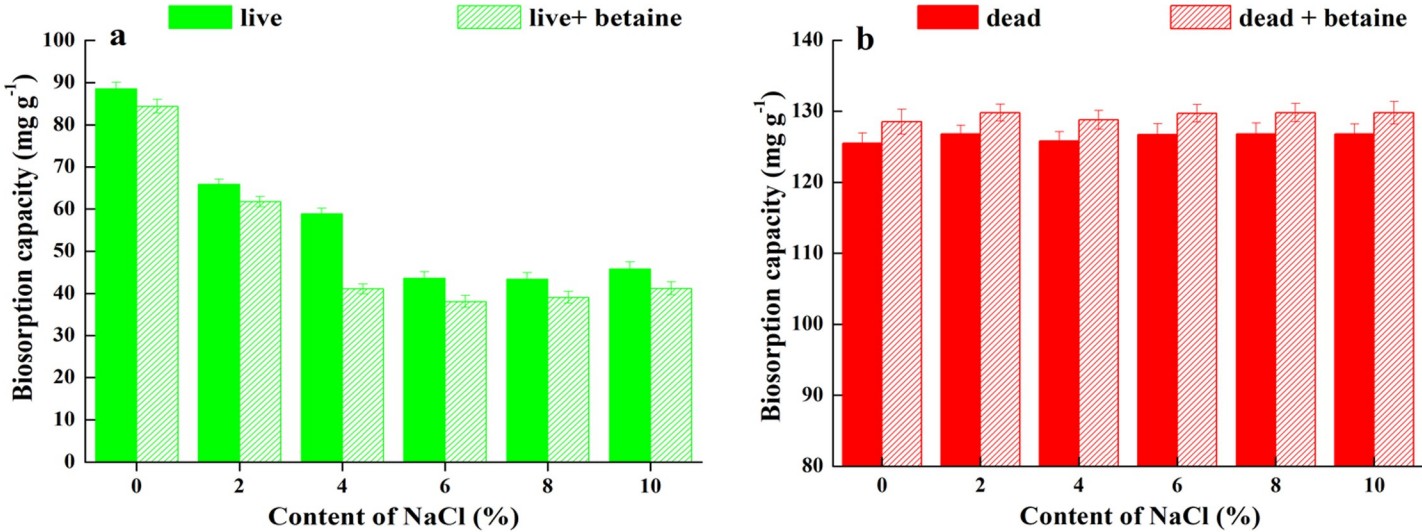

**Fig 4.** Biosorption capacity of Pb$^{2+}$ by adding 150 mg L$^{-1}$ betaine on solution of live(a) and dead(b) biosorbents over NaCl ranging from 0–10%.

From Fig 5A and 5B and S6 Table, the $R^2$ value obtained was enough large, indicating that the experimental $q_e$ value is very close to the calculated value obtained from the linear graph.

Secondly, the pseudo second-order dynamics model is applied to the fitting of experimental data as shown in the following equation:

$$\frac{t}{q_t} = \frac{1}{k_2 q_e^2} + \frac{t}{q_e}$$

Fig 5C and 5D show a plot of the linear relationship between $t\ qt^{-1}$ and $t$. S7 Table summarizes the calculated and theoretical $q_e$ values, the correlation coefficient values ($R^2$) and the pseudo second-order rate constants ($k_2$). The theoretical $q_e$ value assumes an equilibrium concentration of live and dead biosorbents after removal of 100% Pb$^{2+}$. The theoretical value is consistent with the calculated $q_e$ value, showing a fairly large linear relationshipwith $R^2$ above 0.99. Therefore, the biosorption kinetics of live and dead biosorbents follows a pseudo second order model. It indicates that it belongs to the chemisorption process and the rate limiting step is a site of chemisorption rather than membrane transport of metal ions [46].

This study also fully considered the intraparticle diffusion kinetics model as a mechanism of biosorption. The amount of biosorption is almost proportional to $t^{0.5}$. According to the following Weber-Morris's equation [47]:

$$q_t = k_{di}\sqrt{t} + C_i \tag{8}$$

where $k_{di}$ is the rate constant of intra-particle diffusion (mg g$^{-1}$ h$^{0.5}$), calculated from the slope of the straight line of $q_t$ versus t$^{0.5}$. $C_i$ is the value of intercept of stage $i$, giving an idea about the boundary layer thickness. The above conditions indicate that the larger the intercept, the larger the boundary layer effect.

Fig 5E shows the intraparticle diffusion model simulating the biosorption of Pb$^{2+}$ by live and dead biosorbents. Three steps in the biosorption process can be observed from the figure. First, metal ion rapid membrane transport to live and dead biosorbents are completed in the first 7.5 minutes. We can think of the driving force of diffusion due to the high concentration of initial heavy metal ions. It can be clearly observed that the membrane transfer rate of Pb$^{2+}$ adsorbed by dead biosorbents in the first stage is higher than live biosorbents (S8 Table). The

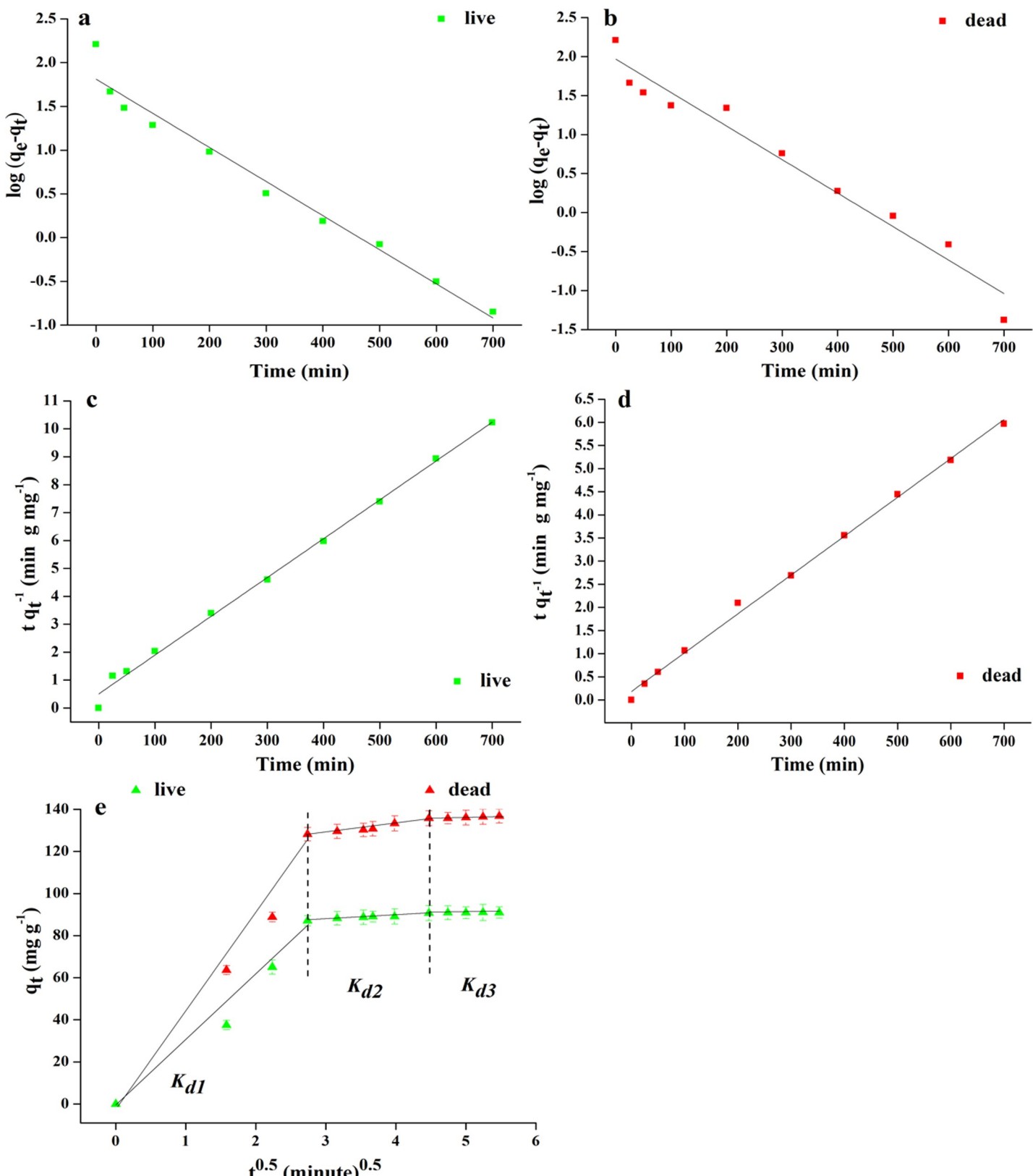

**Fig 5.** (a) Pseudo-first-order kinetics (live biosorbents) (b) Pseudo-first-order kinetics (dead biosorbents) (c) Second-order kinetics (live biosorbents) (d) Second-order kinetics (dead biosorbents) (e) Intra-particle diffusion kinetics for biosorption of heavy metal ions onto the live and dead biosorbents(T = 20°C; biosorbent dose: 0.5 g L$^{-1}$; pH 5; agitation speed: 200 rpm min$^{-1}$).

second step in intraparticle diffusion is a stepwise biosorption phase and it is the rate limiting step throughout the process. This stage may be due to the interaction between the metal ions and the functional groups on the surface of the biosorbent and precipitation conversion occurs within the biosorbent during this process [47]. The slower biosorption in the last step is due to the residual low concentration of heavy metal ions that are not absorbed in the solution [48–49]. The rate parameters $k_{d1}$ and $k_{d2}$ of dead biosorbents adsorbing $Pb^{2+}$ are larger than live biosorbents. We speculate that the reason may be that the surface complexing of dead biosorbents and the precipitation of heavy metals are better than live biosorbents [50]. From the results $k_{d1} > k_{d2} > k_{d3}$, it can be clearly understood that the concentration of unadsorbed metal ions in the solution gradually decreases.

## Desorption efficiency and recycling of biosorbents

Herein we used HCl for pH adjustment. The desorption efficiency of live and dead biosorbents was very low when pH was >5.0 (S5A Fig). In case of live biosorbents, more and more metal ions were released into the supernatant as the desorption process progressed (pH 4–2). When pH decreased to 1.0, the desorption efficiency of dead and live biosorbents increased to 93.5% and 72.7%, respectively.

Moreover, five cycles of biosorption–desorption experiments were performed (S5B Fig). In comparison with the first cycle, the biosorption capacity at the fifth cycle decreased from 114.2 to 79.9 mg/g for dead biosorbents and from 62.6 to 38.8 mg/g for live biosorbents. The recycling efficiency was 70% and 61.9% for dead and live biosorbents, respectively. After five recycling experiments, live and dead biosorbents still showed a strong biosorption capacity, indicating that biosorbents can be applied to the actual metal solution biosorption.

## Effects of coexisting ions on $Pb^{2+}$ biosorption

As shown in S6 Fig, the removal rate of $Pb^{2+}$ was lowered due to the presence of metal ions. For live biosorbents, the removal rate of $Pb^{2+}$ was significantly reduced when only one competitive metal ion was present, and the effect of $Cu^{2+}$ on the removal rate was the strongest. The combination of $Cd^{2+} + Cu^{2+}$ and $Ni^+ + Cd^{2+}$ resulted in a much lower removal rate of $Pb^{2+}$, while the synergistic effect of $Ni^+$ and $Cu^{2+}$ was not as powerful. In the presence of four metal ions, the removal rate decreased from 70.2% to 31.7%. Further, in the presence of a competing ion, dead biosorbents removed less $Pb^{2+}$, and the effect of $Cu^{2+}$ was the greatest, as stated above. In addition, the presence of $Cd^{2+}$ and $Cu^{2+}$ significantly reduced the removal rate of $Pb^{2+}$ in comparison to the presence of the groups 5–6. When all four metal ions coexisted, the removal rate of $Pb^{2+}$ from dead biosorbents decreased from 85.2% to 33.3%. Dead biosorbents reduced $Pb^{2+}$ removal rate to a greater extent when all competing ions were present. Herein the order of influence of single metal ions on $Pb^{2+}$ adsorption was $Cu^{2+} > Cd^{2+} > Ni^+$.

## FT-IR analysis

The FT-IR spectrum of native and $Pb^{2+}$-loaded live and dead biosorbents is shown in Fig 6; functional groups that may be involved in the biosorption process were analyzed. In accordance with the literature [51–52], the FT-IR spectrum of native and dead biosorbents showed a broad and strong peak at 3300.06 $cm^{-1}$, indicating the presence of a hydroxyl (-OH) or amine (-NH) group. The peaks at 2929.75 and 2863.31 $cm^{-1}$ could be due to the stretching vibration of the $-CH_2$ group, and those at 1664.49 and 1542.98 $cm^{-1}$ could be attributed to the stretching vibration of the -NH and -CN groups, respectively. Moreover, the peak at 1404.12 $cm^{-1}$ (mainly C–N stretch) could be attributed to the amide I, II, and III bonds of the protein

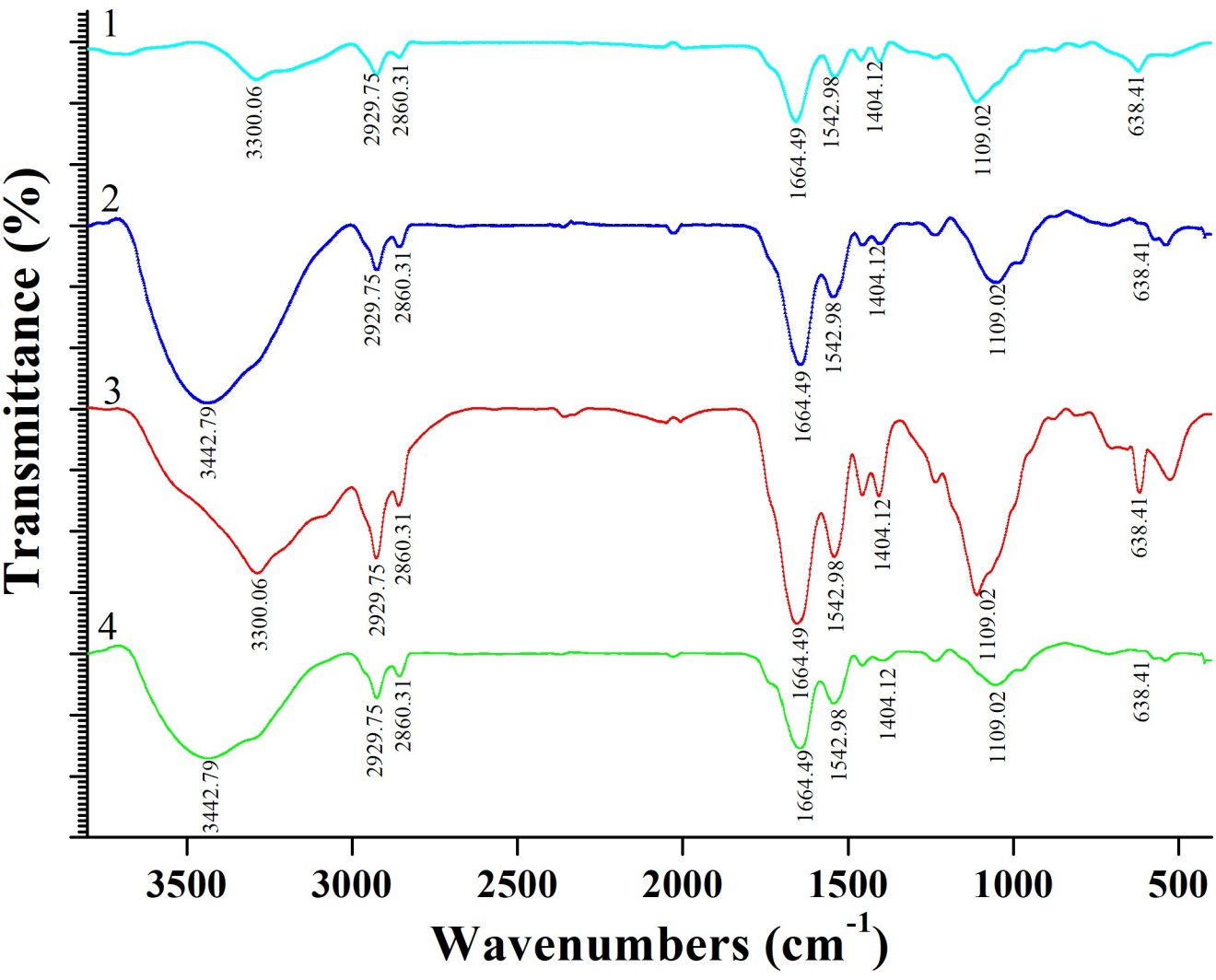

**Fig 6. FT-IR spectrum of live and dead biosorbents loaded with and without Pb$^{2+}$ (1: Native live biosorbents; 2: Pb$^{2+}$ loaded live biosorbents; 3: Nativedead biosorbents; 4: Pb$^{2+}$ loaded dead biosorbents; Pb$^{2+}$: 200 mg L$^{-1}$).**

fraction. The medium strength bonding between live and dead biosorbents of 1109.02 cm$^{-1}$ could be due to the C–N stretching vibration of the amide bond or the C–O stretching of alcohol and carboxylic acid. A specific absorption band of the aromatic structure was also observed at 638.41 cm$^{-1}$.

## SEM characterization of live and dead biosorbents

Fig 7 shows SEM images of the biosorbents before and after Pb$^{2+}$ biosorption. Before biosorption, the biosorbents showed a smooth surface and were observed to grow in a strip shape (Fig 7A). However, changes in bioactive biosorption occurred in both live and dead biosorbents, such as deformation of some cells, leading to rough and ruptured surfaces in case of live biosorbents (Fig 7C); in addition, biosorption of a small amount of Pb$^{2+}$ can be observed. Further, dead biosorbents (Fig 7B) were found to adsorb a large amount of heavy metals on the surface and showed no significant change in surface morphology. Energy-dispersive X-ray analysis (Fig 7, S9 and S11 Tables) confirmed Pb$^{2+}$ biosorption in biosorbents; the main elements of

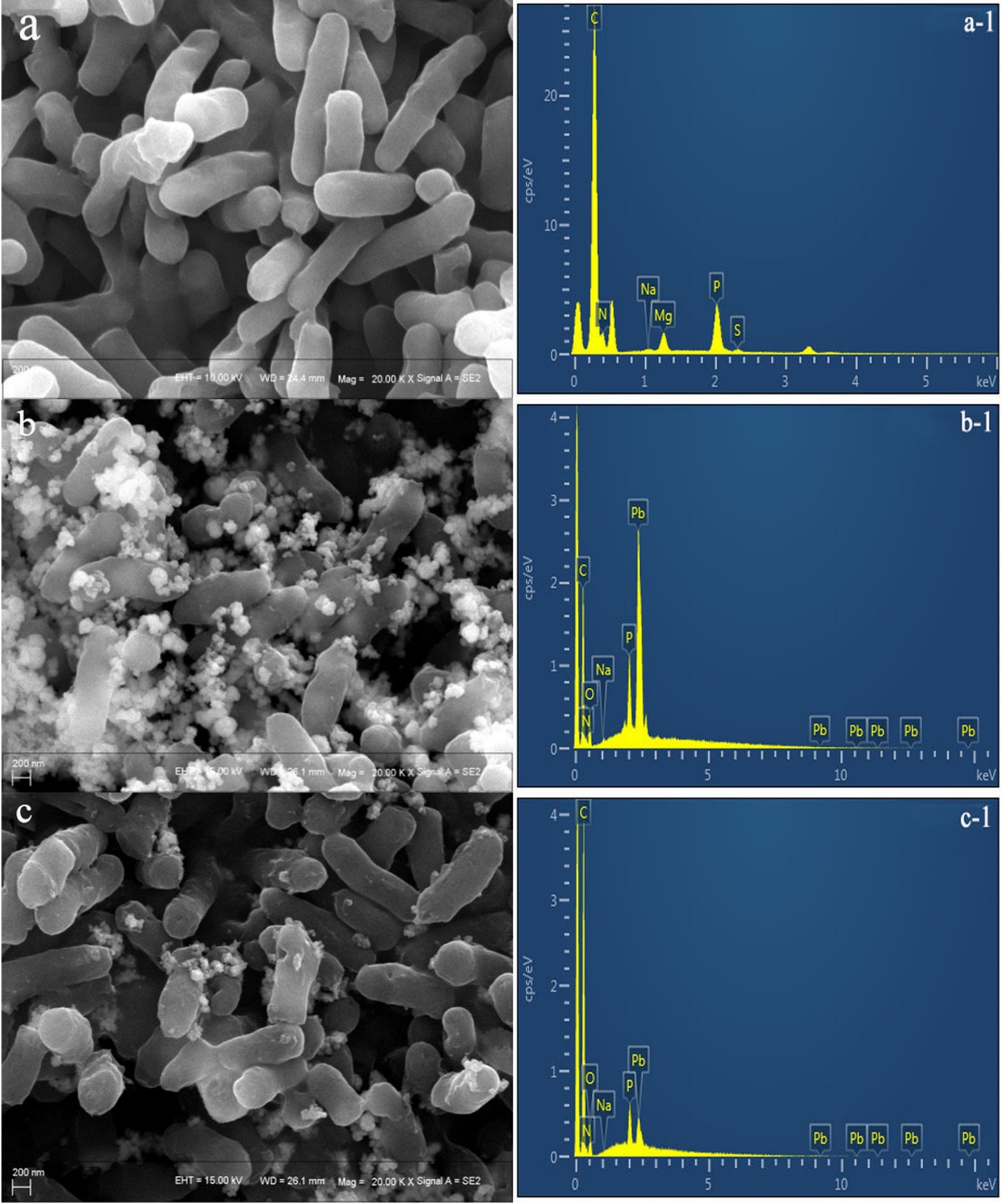

**Fig 7.** SEM image and EDX of biosorption $Pb^{2+}$ by live and dead biosorbents (a: Native live biosorbents; a-1: EDX for native live biosorbents; b: $Pb^{2+}$ loaded dead biosorbents; b-1: EDX for $Pb^{2+}$ loaded dead biosorbents; c: $Pb^{2+}$ loaded live biosorbents; c-1: EDX for $Pb^{2+}$ loaded live biosorbents; $Pb^{2+}$: 200 mg $L^{-1}$).

dead biosorbents after $Pb^{2+}$ biosorption were C (35.66%), O (8.55%), P (4.54%), and Pb (49.23%), and those of live biosorbents after $Pb^{2+}$ biosorption were C (72.67%), O (14.12%), P (4.11%), and Pb (9.08%). Based on these results, the amount of heavy metals adsorbed by dead biosorbents was indeed much higher than that by live biosorbents.

### TEM

Electron microscopy was performed on samples before and after $Pb^{2+}$ biosorption on live and dead biosorbents. An electron micrograph of the control biosorbent (S7A Fig, S12 Table) showed a normal appearance. Electron micrographs of live and dead biosorbents with bio-sorbed $Pb^{2+}$ (S7B and S7C Fig) showed different states. Dead biosorbents showed intact cell morphology after $Pb^{2+}$ biosorption, but many white shadows were visible in the cells, which could be due to $Pb^{2+}$ accumulation. Energy-dispersive X-ray analysis validated this observation (S7B-1 Fig, S13 Table). In contrast, the surface of live biosorbents after heavy metal biosorp-tion was severely damaged, and heavy metal accumulation was clearly visible in the cells. Energy-dispersive X-ray analysis also indicated that $Pb^{2+}$ was indeed present in the cells, with the content being 1.64% (S7C-1 Fig, S14 Table). These data indicated that dead biosorbents are more favorable than live biosorbents for $Pb^{2+}$ biosorption.

## Discussion

Studying the tolerance of strains to different heavy metals is the primary prerequisite for exploring biosorption. *Rhodococcus* sp. RS67 is reportedly resistant to $Pb^{2+}$, with the minimum inhibitory concentration (MIC) being 103.6 mg $L^{-1}$ [53]; similarly, for KF790905.1, the MIC is >330 mg $L^{-1}$ [54]. In the current study, we used *Rhodococcus* sp. HX-2; this strain has a good tolerance to $Pb^{2+}$, with the MIC being 500 mg $L^{-1}$, which is much higher than above reports. This also indicates its potential of $Pb^{2+}$ biosorption.

The four conditions, namely biosorption dose, pH, temperature, and contact time, are indispensable for the optimization of $Pb^{2+}$ biosorption. Early research by Li et al. [37] reported that biosorbent dose is also an important parameter influencing removal rate and biosorption capacity. Obviously, the number of binding sites increase with an increase in biosorbent dose, consequently increasing the removal rate of $Pb^{2+}$. However, excessive biosorbents are unneces-sary for effective biosorption at a certain concentration of metal ions, as biosorption efficiency begins to decrease as the concentration of metal ions decrease. In the present study, we observed that biosorption capacities for both live and dead biosorbents began to decrease as the pH value increased beyond 5.0. It has been reported that when pH increases to 5.0, more negatively charged functional groups, such as carboxyl, hydroxyl, and phosphate groups, are exposed. Therefore, the number of attraction sites increases and more positively charged ions can be absorbed, thereby increasing biosorption capacity [55]. The relatively low biosorption capacity at lower pH in this study can be attributed to protonation of the active site, resulting in the occupancy of the competitive binding site between $H^+$ and $Pb^{2+}$ [56]. The decrease in biosorption at higher pH may be attributed to the formation of $Pb(OH)_2$ by $Pb^{2+}$ and hydrox-ide in the solution, resulting in a decrease in the number of adsorbable metal ions. Therefore, we speculate that pH dependence of a biosorbent on metal absorption is not only the proton-ation of surface functional groups on the cell wall but also intracellular chemicals and metal forms in the solution [56]. Dead biosorbents do not affect biosorption capacity with a change in temperature; this could be as they are insensitive to changes in temperature and do not alter the morphology and functional groups of the biosorbent. Live biosorbents can be affected by a change in temperature, thereby increasing or decreasing biosorption capacity. This could be because higher temperatures are more favorable for the biosorption of $Pb^{2+}$ by live

biosorbents. When the temperature exceeds the optimum value, the biosorption capacity of live biosorbents begins to decline. As per published literature, the two stages of biosorption include the initial rapid uptake phase, followed by the slow uptake phase of intracellular diffusion due to higher metal ion concentrations. The first stage involves the rapid transport of metal ions into live and dead biosorbents (1–3 h) [57]. In this study, in the presence of live and dead biosorbents, the contact time reached equilibrium in 7.5 min, which is shorter than the biosorption time reported by Luna et al. [57]. This indicates that live and dead biosorbents of HX-2 can biosorb heavy metals more rapidly.

We also explored the effects of different concentrations of $Pb^{2+}$ on biosorption. However, when $Pb^{2+}$ concentration exceeded 150 mg $L^{-1}$, the difference between the equilibrium biosorption capacity of live and dead biosorbents seemed to increase with an increase in the initial concentration. When $Pb^{2+}$ concentration was increased to 200 mg $L^{-1}$, the biosorption capacity of live and dead biosorbents no longer increased. The equilibrium biosorption capacity of live biosorbents was 88.74 mg $g^{-1}$ (0.428 mmol $g^{-1}$) $Pb^{2+}$, while dead biosorbents could absorb 125.5 mg $g^{-1}$ (0.606 mmol $g^{-1}$) $Pb^{2+}$. Betaine is an alkaloid that plays a significant role in maintaining cell osmotic pressure and alleviating salt stress. It has good solubility, no static charge, and no effect on most enzymes and biomacromolecules in many organisms at high concentrations [58]. Moreover, betaine can relieve high salt concentration toxicity, making it a common and highly effective osmoprotectant in prokaryotes [42]. The addition of betaine did not appear to substantially affect the biosorption capacity of live or dead biosorbents; this could be because betaine is only related to the tolerance of the strain to NaCl, rather than to the biosorption of heavy metals.

Reddy et al. reported that the adsorption kinetic data were best described by the pseudo-second-order model ($Pb^{2+}$) [59]. Abu-Danso et al. also revealed that the adsorption kinetics of Pb(II) and Cd(II) onto a clay–cellulose biocomposite was well described by the pseudo-second-order kinetic model [60]. Similarly, our experimental results proved that the biosorption kinetics of live and dead biosorbents followed the pseudo-second-order model. Both biosorbent regeneration and metal ion recovery are closely related to the desorption process. The desorption process also indicates the degree of binding of heavy metal ions to biomass. The stronger the degree of binding, the lesser is the amount of heavy metal ions released into the supernatant [61–62]. In addition, dead biosorbents have a higher desorption efficiency than live biosorbents, and the binding strength of surface metal ions ($Pb^{2+}$) to dead biosorbents appears to be slightly weaker than that of live biosorbents. Our results showed that the desorption process is related to pH of the solution and type of biosorbent. The recyclability of biosorbents is also an important condition affecting biosorption. Kwak et al. reported that the successive recycling of the adsorption–desorption ($Cr^{6+}$) process was stable for more than five cycles, and the recycling efficiency was 70% [63]. Pan et al. reported that $Eu^{3+}$ could be well desorbed by HCl or EDTA solution and that the bacterial strain exhibited good regeneration and reusability [64]. Furthermore, Das et al. showed that the biosorbent in their study could be consistently reused for up to five cycles with a minor metal leaching of 0.92% [65]. In our study, after five recycling experiments, the recycling efficiency was 70% and 61.9% for dead and live biosorbents, respectively, indicating that the biosorbents could be well regenerated and reused. Pertaining to the effect of coexisting ions on $Pb^{2+}$ biosorption, there are three possible interactive effects of a mixture of metal ions, namely synergism [66], antagonism [67], and non-interaction [68]. The reason for the removal of a metal ion in the presence of other metal or inorganic ions ($Na^+$, $Ca^{2+}$) is that the metal binding sites on the biosorbent are limited. Therefore, when three metal ions coexisted, the removal rate of $Pb^{2+}$ by live and dead biosorbents was the lowest, and the order of influence of single metal ions on $Pb^{2+}$ adsorption was $Cu^{2+} > Cd^{2+} > Ni^+$.

FT-IR spectrum showed that $Pb^{2+}$-loaded live biosorbents did not change much in comparison to native live biosorbents. Only the peak of -OH or -NH vibration was observed to shift to 3442.79 $cm^{-1}$. This indicates that the above groups can participate in the $Pb^{2+}$ biosorption process. Masoumi et al. reported that *Curtobacterium* sp. FM01 could be a promising candidate with a capacity to remove Ni (II) and Pb (II) from aqueous solutions [69], and their findings are similar to our results. As for dead biosorbents, FT-IR analysis indicated that the hydroxyl (-OH), alkyl ($-CH_2$), amino (-NH), nitrile (-CH), and aromatic ($-C_6H_5$) groups were involved in $Pb^{2+}$ biosorption. Thus, in comparison with live biosorbents, more functional groups seem to be involved in the $Pb^{2+}$ biosorption process of dead biosorbents. Many authors have also reported surface morphological changes in bacteria after the adsorption of $Pb^{2+}$ [70–71]. The cell surface after $Pb^{2+}$ biosorption by living bacteria can show different degrees of damage; on the other hand, dead bacterial cells remain unaffected by metal ions. Our results are consistent with those of previous studies. TEM further proved that $Pb^{2+}$ was transferred inside the cells by live and dead biosorbents, increasing external biosorption sites, which is more conducive to large biosorption of metal ions.

In recent years, many studies have reported the adsorption of heavy metals by functional materials or bacteria as adsorbents. Tripathi et al. reported that carbon nanodot material was easily modified through simple oxidative treatment with nitric acid, and thus, they proposed that this material could be promising for biosensing and drug delivery applications [72]. Graphene aerogel, synthesized from crude biomass, has been proposed to be an ecofriendly cell growth promoter and a highly efficient adsorbent for histamine, which is a food toxicant, removal from red wine [73]. Das et al. used nitrogen-doped soluble graphene nanosheets as multifunctional materials for photocatalytic, sensing, and adsorption applications [74]. Moreover, Sulaymon et al. reported that dead and live anaerobic biomass could be used for biosorption of Pb (II) ions from synthetic wastewater, and the uptake capacity was 51.56 mg $g^{-1}$ [75]. Sulaymon et al. also performed a comparison between live and dead microorganisms for removing phenol and lead from aqueous solutions; the maximum loading capacity for lead was found to be 36.7888 and 89.8783 mg $g^{-1}$, respectively [76]. Batch biosorption of Pb (II) using both live and dead biomass of the marine bacterium *Bacillus xiamenensis* was investigated by Mohapatra et al.; the maximum Pb (II) uptake of 216.75 and 207.4 mg $g^{-1}$ biomass was obtained with live and dead biomass, respectively [77]. In our study, live and dead biosorbents of *Rhodococcus* sp. HX-2 were used to biosorb $Pb^{2+}$ from aqueous solutions. The maximum biosorption capacity was 88.74 mg $g^{-1}$ (0.428 mmol $g^{-1}$) for live biosorbents and 125.5 mg $g^{-1}$ (0.606 mmol $g^{-1}$) for dead biosorbents. Only a few studies have explored $Pb^{2+}$ biosorption by biosorbents of *Rhodococcus* sp., proving the importance and relevance of the present study.

In conclusion, we first used the response surface methodology to optimize the biosorption of $Pb^{2+}$ by live and dead biosorbents. The effects of external conditions, isotherm and kinetic models, desorption efficiency, and coexisting ions on $Pb^{2+}$ biosorption were explored. Live and dead biosorbents were characterized both before and after $Pb^{2+}$ biosorption by FT-IR, SEM, and TEM. The maximum biosorption capacities were 88.74 mg $g^{-1}$ (0.428 mmol $g^{-1}$) and 125.5 mg $g^{-1}$ (0.606 mmol $g^{-1}$) for live and dead biosorbents, respectively. At an initial $Pb^{2+}$ concentration of 200 mg $L^{-1}$, the biosorption data of live and dead biosorbents were suitable for the Freundlich model. With higher $R^2$ values and more accurate $q_e$ prediction, pseudo-second-order equations can describe better biosorption kinetics than pseudo-first-order equations. Desorption experiments indicated that dead biosorbents have more potential sites than live biosorbents. Further, FT-IR analysis showed that dead biosorbent had more functional groups than live biosorbents to participate in the biosorption process (-OH, $-CH_2$, -NH, -CH, $-C_6H_5$). SEM showed that $Pb^{2+}$ was heavily adsorbed onto the surface of dead

biosorbents, and TEM revealed that live and dead biosorbents transferred biosorbed $Pb^{2+}$ inside the cell. Considering these findings, dead biosorbents seem to be a more effective than live biosorbents, justifying their application for wastewater treatment.

## Supporting information

**S1 Table. Variables and levels for Box-Behnken design.**
(PDF)

**S2 Table. Box Behnken design matrix and response values.**
(PDF)

**S3 Table. Analysis of variance for the response surface quadratic model.**
(PDF)

**S4 Table. The predicted and experimental value of response under optimum conditions.**
(PDF)

**S5 Table. Langmuir adsorption isotherms and Freundlich adsorption isotherms for $Pb^{2+}$ using live and dead biosorbents.**
(PDF)

**S6 Table. Pseudo-first-order adsorption kinetic constants of the live and dead biosorbents.**
(PDF)

**S7 Table. Pseudo-second-order adsorption kinetic constants of the live and dead biosorbents.**
(PDF)

**S8 Table. Intra-particle diffusion model constants and correlation coefficients forbiosorption of metal ions on the live and dead biosorbents.**
(PDF)

**S9 Table. EDX (SEM) analysis for natural biosorbents.**
(PDF)

**S10 Table. EDX (SEM)analysis for $Pb^{2+}$ loaded dead biosorbents.**
(PDF)

**S11 Table. EDX (SEM) analysis for $Pb^{2+}$ loaded live biosorbents.**
(PDF)

**S12 Table. EDX (TEM) analysis for natural biosorbents.**
(PDF)

**S13 Table. EDX (TEM)analysis for $Pb^{2+}$ loaded dead biosorbents.**
(PDF)

**S14 Table. EDX (TEM) for $Pb^{2+}$ loaded live biosorbents.**
(PDF)

**S1 Fig. Predict values vs. actual values of the response.**
(PDF)

**S2 Fig. Contour plots and three-dimensional response plots for the response.**
(PDF)

**S3 Fig.** Langmuir adsorption isotherms for $Pb^{2+}$ using live (a) and dead biosorbents (b).
(PDF)

**S4 Fig.** Freundlich adsorption isotherms for $Pb^{2+}$ using live (a) and dead biosorbents (b).
(PDF)

**S5 Fig.** Effect of pH on desorption efficiency of $Pb^{2+}$ from live and dead biosorbents (a) and
the recycling experiments (b).
(PDF)

**S6 Fig. Effect of competing ions on removal efficiency of $Pb^{2+}$ by the live and dead biosorbents.**
(PDF)

**S7 Fig. TEM image and EDX of biosorption $Pb^{2+}$ by live and dead biosorbents.**
(PDF)

**S1 Equation. Pseudo-first-order equation.**
(PDF)

## Acknowledgments

The authors would like to thank Yi Qi and Jinbiao Liu for invaluable technical assistance in
laboratory facility.

## Author Contributions

**Investigation:** Jiachang Cao.

**Methodology:** Jiachang Cao, Dahui Li, Zhixia Zhang.

**Project administration:** Dahui Li, Zhixia Zhang.

**Resources:** Yue Qiao.

**Software:** Yue Qiao.

**Supervision:** Jialin Zhao.

**Validation:** Jialin Zhao.

**Writing – original draft:** Xin Hu, Lei Huang.

**Writing – review & editing:** Xin Hu, Hanyu Yang, Lei Huang.

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
