## [Decision Letter · Decision Letter 0]

25 Oct 2019

PONE-D-19-26926

Biosorption of Pb2+ from aqueous solution by live and dead biosorbents of hydrocarbon-degrading strain Rhodococcus HX-2

PLOS ONE

Dear Mr. Huang,

Thank you for submitting your manuscript to PLOS ONE. After careful consideration, we feel that it has merit but does not fully meet PLOS ONE’s publication criteria as it currently stands. Therefore, we invite you to submit a revised version of the manuscript that CAREFULLY addresses the points raised during the review process.

We would appreciate receiving your revised manuscript by Dec 09 2019 11:59PM. To enhance the reproducibility of your results, we recommend that if applicable you deposit your laboratory protocols in protocols.io, where a protocol can be assigned its own identifier (DOI) such that it can be cited independently in the future. For instructions see: http://journals.plos.org/plosone/s/submission-guidelines#loc-laboratory-protocols

We look forward to receiving your revised manuscript.

Kind regards,

Yogendra Kumar Mishra, Ph. D.

Academic Editor

PLOS ONE

Journal Requirements:

2. In your Methods section, please provide additional location information of the soil collection site, including geographic coordinates for the data set if available.

3. In your Methods section, please provide additional information regarding the permits you obtained for the work. Please ensure you have included the full name of the authority that approved the collection site access and, if no permits were required, a brief statement explaining why.

Additional Editor Comments (if provided):

Reviewers' comments:

Reviewer's Responses to Questions

**Comments to the Author**

1. Is the manuscript technically sound, and do the data support the conclusions?

Reviewer #1: Yes

Reviewer #2: Yes

2. Has the statistical analysis been performed appropriately and rigorously? 

Reviewer #1: Yes

Reviewer #2: Yes

3. Have the authors made all data underlying the findings in their manuscript fully available?

Reviewer #1: Yes

Reviewer #2: Yes

4. Is the manuscript presented in an intelligible fashion and written in standard English?

Reviewer #1: Yes

Reviewer #2: No

5. Review Comments to the Author

Reviewer #1: Present manuscript reports the biosorption of Pb(II) using live and dead Rhodococcus as bio-adsorbents. Further the authors reported the kinetics of biosorption. The manuscript is well written and can be accepted after major revisions due to the following ambiguous discussions.

1. The recycling experiments for the biosorption of Pb should be supplemented to ensure the stability.

2. Does Pb harm bacterial colonies?

3. The whole expression of the manuscript should be carefully modified to be scientific and vigorous.

4. A large body of literatures on related research works in toxic metal adsorption are missed.

Journal of Hazardous Materials, 2019, DOI: 10.1016/j.jhazmat.2019.120871. ACS

New J. Chem., 2019,43, 14575-14583. Applied Materials and Interfaces, 2019, 11, 18165-18177. Journal of Nanomaterials, 2017, 2017, 7029731.

Reviewer #2: In Manuscript PONE-D-19-26926, live and dead biosorbents of hydrocarbon-degrading strain Rhodococcus HX-2 were used and compared for the biosorption of Pb2+ from aqueous solution. The dosage of biosorbents, initial metal concentration, pH, and contact time were compared and analysed. In addition, different isothermal and kinetic models were described for this mechanism of biosorption.

This work could be interesting for future research designs in water treatment/management, while using live biosorbents. This manuscript can be considered for further publication after some minor revisions.

1. More literature based on live and/or dead biosorbents for the sorption of Pb2+ from aqueous solution and comparative discussion regarding to this current study.

2. Figures are not clear, especially Fig. 6. Please provide clear version of the figures.

6. PLOS authors have the option to publish the peer review history of their article (what does this mean?). If published, this will include your full peer review and any attached files.

Reviewer #1: No

Reviewer #2: No

---

## [Author Response · Author response to Decision Letter 0]

19 Nov 2019

List of Responses

Dear Editors and Reviewers:

Thank you for your letter and for the reviewers’ comments concerning our manuscript entitled “Biosorption of Pb2+ from aqueous solution by live and dead biosorbents of hydrocarbon-degrading strain Rhodococcus HX-2” (ID: PONE-D-19-26926). Those comments are all valuable and very helpful for revising and improving our paper, as well as the important guiding significance to our researches. We have studied comments carefully and have made correction which we hope meet with approval. Revised portion are marked in yellow in the paper. 

Deer editor, we have completed the modification of the article format. Please see the manuscript and response to reviewers for details.

---

## [Decision Letter · Decision Letter 1]

3 Dec 2019

Pb2+ biosorption from aqueous solutions by live and dead biosorbents of the hydrocarbon-degrading strain Rhodococcus sp. HX-2

PONE-D-19-26926R1

Dear Dr. Huang,

We are pleased to inform you that your manuscript has been judged scientifically suitable for publication and will be formally accepted for publication once it complies with all outstanding technical requirements.

With kind regards,

Yogendra Kumar Mishra, Ph. D.

Academic Editor

PLOS ONE

Additional Editor Comments (optional):

Reviewers' comments:

Reviewer's Responses to Questions

**Comments to the Author**

1. If the authors have adequately addressed your comments raised in a previous round of review and you feel that this manuscript is now acceptable for publication, you may indicate that here to bypass the “Comments to the Author” section, enter your conflict of interest statement in the “Confidential to Editor” section, and submit your "Accept" recommendation.

Reviewer #1: All comments have been addressed

Reviewer #2: All comments have been addressed

2. Is the manuscript technically sound, and do the data support the conclusions?

Reviewer #1: Yes

Reviewer #2: Yes

3. Has the statistical analysis been performed appropriately and rigorously? 

Reviewer #1: Yes

Reviewer #2: Yes

4. Have the authors made all data underlying the findings in their manuscript fully available?

Reviewer #1: Yes

Reviewer #2: Yes

5. Is the manuscript presented in an intelligible fashion and written in standard English?

Reviewer #1: Yes

Reviewer #2: Yes

6. Review Comments to the Author

Reviewer #1: (No Response)

Reviewer #2: Authors have responded well agaisnt the Reviewer's comments and improved manuscript accordingly. In my opinion, this manuscript can now be accepted for publication.

7. PLOS authors have the option to publish the peer review history of their article (what does this mean?). If published, this will include your full peer review and any attached files.

Reviewer #1: No

Reviewer #2: No

---

## [Editor Report · Acceptance letter]

21 Jan 2020

PONE-D-19-26926R1 

Pb^2+^ biosorption from aqueous solutions by live and dead biosorbents of the hydrocarbon-degrading strain *Rhodococcus* sp. HX-2 

Dear Dr. Huang:

I am pleased to inform you that your manuscript has been deemed suitable for publication in PLOS ONE. Congratulations! Your manuscript is now with our production department. 

With kind regards,

on behalf of

Dr. Yogendra Kumar Mishra 

Academic Editor

PLOS ONE